# Ultrasonography in the Diagnosis of Adnexal Lesions: The Role of Texture Analysis

**DOI:** 10.3390/diagnostics11050812

**Published:** 2021-04-29

**Authors:** Paul-Andrei Ștefan, Roxana-Adelina Lupean, Carmen Mihaela Mihu, Andrei Lebovici, Mihaela Daniela Oancea, Liviu Hîțu, Daniel Duma, Csaba Csutak

**Affiliations:** 1Anatomy and Embryology, Morphological Sciences Department, “Iuliu Hațieganu” University of Medicine and Pharmacy, Victor Babes Street 8, 400012 Cluj-Napoca, Romania; stefan_paul@ymail.com; 2Radiology and Imaging Department, County Emergency Hospital, Clinicilor Street 5, 400006 Cluj-Napoca, Romania; carmenmihu@umfcluj.ro (C.M.M.); andrei1079@yahoo.com (A.L.); daniel_duma1992@yahoo.com (D.D.); csutakcsaba@yahoo.com (C.C.); 3Histology, Morphological Sciences Department, “Iuliu Hațieganu” University of Medicine and Pharmacy, Louis Pasteur Street 4, 400349 Cluj-Napoca, Romania; 4Obstetrics and Gynecology Clinic “Dominic Stanca”, County Emergency Hospital, 21 Decembrie 1989 Boulevard 55, 400094 Cluj-Napoca, Romania; oancea.mihaela@umfcluj.ro; 5Radiology, Surgical Specialties Department, “Iuliu Hațieganu” University of Medicine and Pharmacy, Clinicilor Street 3–5, 400006 Cluj-Napoca, Romania; 6Obstetrics and Gynecology Clinic II, Mother and Child Department, “Iuliu Hațieganu” University of Medicine and Pharmacy, 21 Decembrie 1989 Boulevard 55, 400094 Cluj-Napoca, Romania; 7Doctoral School, Iuliu Hațieganu University of Medicine and Pharmacy, 400012 Cluj-Napoca, Romania; Liviu.Hitu@umfcluj.ro

**Keywords:** computer-aided diagnosis, ovarian cyst, ovarian tumor, texture analysis, ultrasonography

## Abstract

The classic ultrasonographic differentiation between benign and malignant adnexal masses encounters several limitations. Ultrasonography-based texture analysis (USTA) offers a new perspective, but its role has been incompletely evaluated. This study aimed to further investigate USTA’s capacity in differentiating benign from malignant adnexal tumors, as well as comparing the workflow and the results with previously-published research. A total of 123 adnexal lesions (benign, 88; malignant, 35) were retrospectively included. The USTA was performed on dedicated software. By applying three reduction techniques, 23 features with the highest discriminatory potential were selected. The features’ ability to identify ovarian malignancies was evaluated through univariate, multivariate, and receiver operating characteristics analyses, and also by the use of the k-nearest neighbor (KNN) classifier. Three parameters were independent predictors for ovarian neoplasms (sum variance, and two variations of the sum of squares). Benign and malignant lesions were differentiated with 90.48% sensitivity and 93.1% specificity by the prediction model (which included the three independent predictors), and with 71.43–80% sensitivity and 87.5–89.77% specificity by the KNN classifier. The USTA shows statistically significant differences between the textures of the two groups, but it is unclear whether the parameters can reflect the true histopathological characteristics of adnexal lesions.

## 1. Introduction

Transvaginal ultrasonography (TVUS) is the first-choice technique used in the characterization of a suspicious adnexal mass [1]. One of the most important goals of imaging when assessing these lesions is differentiating between malignant and benign ones [2], which has a high impact on subsequent medical and surgical management [3].

Several attributes that are considered to advocate for malignancy have been described (such as multilocularity, irregular and thickened cystic septations, and internal vegetations) [3], and scoring systems have been developed [4] in the attempt of improving ovarian cancers’ detection rates. Nevertheless, the neoplastic morphological features are inconsistent and not pathognomonic, since some mixed-type benign tumors can sometimes mimic the aspect of a neoplastic mass [3,5]. Often, changes between imaging features corresponding to benign and neoplastic adnexal lesions are subtle and overlapping, so that even experts are prone to give the wrong interpretation [6,7]. Also, the multitude of ultrasound (US) based scoring systems may create confusion, especially since their embedded parameters are variable, complex, and often with arbitrarily defined importance [4,8,9]. Moreover, the interpretation of TVUS images is subjective and observer-dependent [10].

Over the past few years, computer-aided diagnosis (CAD) techniques have been developed to reduce these limitations and provide more confidence in the US diagnosis of ovarian neoplasms [6]. All of the previously developed ultrasound CAD methods rely on texture analysis [7,11,12,13,14,15,16] and attempt to develop classifiers that automatically recognize the presence of the disease based on the detected grayscale variations within the US images [7]. Texture analysis (TA) is a technique that involves the extraction and processing of parameters that reflect the pixel intensity and variation patterns, thus providing a quantitative and detailed description of the image content [17,18]. The basic principle of ultrasonography-based texture analysis (USTA) is that a pathological process that alters the tissue produces a modified US signal, which will in turn give textural features different values from those of the normal structure [19]. Previously published USTA studies [7,11,12,13,14,15,16] that aimed to differentiate benign from malignant adnexal lesions showed the good discriminative abilities of the CAD applications (55–100% accuracy; 55–100% sensitivity; 49–100% specificity). However, due to the pilot nature of these researches, they encounter several pitfalls that include but are not limited to small study populations [7,11,12,13,16], lack of pre-processing methods [15], and the use of few texture classes [15,16], as well as several inconsistencies regarding the statistical analysis and the overall design of the prediction models [7,11,12,13,14,15,16].

This study aimed to perform a detailed texture assessment of benign and malignant adnexal lesions and to further investigate whether previously proposed USTA methods can be reproduced to provide an automated means of differentiation between the two histopathological entities.

## 2. Materials and Methods

### 2.1. Study Group

This Health Insurance Portability and Accountability Act—a compliant, single-institution, retrospective study has been approved by the institutional review board (ethics committee of the “Iuliu Hațieganu” University of Medicine and Pharmacy Cluj-Napoca; registration number: 50; date: 11 March 2019), and informed consent was waived owing to its retrospective nature. From October 2017 to February 2019, a search in the imaging database of our institution was conducted to identify TVUS images corresponding to adnexal lesions. The inclusion criteria were: a lesion with a minimum diameter of at least 20 mm, the availability of conventional B-mode images, lack of imaging artifacts, and the existence of a patient’s serial number (PSN). In total, 413 images belonging to 274 patients were selected. Based on the PSN, the patients’ medical records were retrieved from the archive of our healthcare unit and searched for disease-related data. The exclusion criteria were: no medical data corresponding to the PSN, the absence of a final pathological diagnosis to indicate the benign or malignant nature of the lesions, the pathological analysis performed at more than 30 days after the image acquisition, and no gynecological follow-up.

After applying the inclusion and exclusion criteria, 123 images corresponding to 120 patients were retrieved. From 117 subjects, only one image corresponding to a single lesion was selected. Three subjects had two different pathologically proven lesions, and one image corresponding to each lesion was retrieved (one patient with teratoma and functional cyst, one with endometrioma and functional cyst, and one with endometrioma and hemorrhagic cyst).

According to the patients’ final diagnosis, images were divided into benign (*n* = 88) and malignant (*n* = 35) groups. The benign grop included: functional cysts (*n* = 7), hemorrhagic cysts (*n* = 5), endometriomas (*n* = 28), serous cystadenomas (*n* = 26), mesothelial inclusion cysts (*n* = 2), mucinous cystadenomas (*n* = 6), ovarian abscesses (*n* = 6), oophoritis-related cysts (*n* = 2), and teratomas (*n* = 6). The malignant group included: serous ovarian carcinomas (*n* = 24), endometroid carcinoma of the ovary (*n* = 2), mucinous ovarian carcinomas (*n* = 6), and clear-cell ovarian carcinomas (*n* = 3).

### 2.2. Reference Standard

The pathological examination was performed in the same institution and comprised macroscopic and microscopic evaluation of the lesions. When necessary, the examination was supplemented with immunohistochemical analysis. One to three samples of solid tissues were collected and analyzed microscopically after being stained with hematoxylin and eosin. Of the seven histologically proven functional cysts, two belonged to patients who had other benign lesions and were also included in our study (one with endometrioma and one with teratoma). The other five functional cysts underwent pathological analysis as they were included in the same surgical specimen that was removed for another underlying disease (uterine fibromatosis, *n* = 3; atypical endometrial hyperplasia, *n* = 1; adnexal torsion, *n* = 1). Of the five included hemorrhagic cysts, one belonged to a patient with endometrioma, which was also included in this study. Four hemorrhagic cysts were included in the surgical specimen analyzed for another disease (refractory adenomyosis, *n* = 1; uterine sarcoma, *n* = 1; uterine fibromatosis, *n* = 2).

### 2.3. Image Acquisition and Interpretation

All the examinations were performed by five gynecologists with at least 7 yearsof experience in TVUS on the same unit (Aplio 300, Toshiba Medical Systems, Tokyo, Japan), using a dedicated endovaginal probe (4–10 MHz). Each of the selected examinations were comprised of several images, but only conventional B-mode images were selected. Images were retrieved in digital imaging and communications in medicine (DICOM) format and imported into a dedicated radiology workstation (General Electric, Advantage workstation, 4.7 edition, Waukesha, WI, USA). Images were reviewed by two researchers (one gynecologist and one radiologist, M.D.O. and C.M.M.), who were aware of the subjects’ final diagnostics, pathological findings, and clinical outcomes. When multiple adnexal lesions were observed on the same patient, the TVUS examinations were cross-referenced with the medical data to ensure the selection of only the lesions that were previously documented. Respective lesions were marked, and for every lesion, one image was anonymized and retrieved for subsequent analysis.

### 2.4. Texture Analysis Protocol

The radiomics approach consisted of five steps: image pre-processing, lesion segmentation, feature extraction, feature selection, and prediction.

#### 2.4.1. Image Pre-Processing and Segmentation

Each image was pre-processed in two steps. In the first step, images were converted from DICOM to joint photographic experts group format (JPG). Then, the converted images were imported into a dedicated software (Topaz DeNoise AI, Topaz Labs, TX, USA), where a denoising technique based on convolutional neural networks (CNN) [20] was applied to countermand the negative impact of the speckle noise. After the noise correction, images were reconverted in bitmap format (BMP) and imported into a dedicated texture analysis software (MaZda v5; Institute of Electronics, Technical University of Lodz, Lodz, Poland) [21]. In the second step, within the MaZda software, an image gray-level normalization method (based on the mean and three standard deviations) was applied to reduce the contrast and brightness variations that can affect the true textures of the image [22]. The normalization was automatically performed by the MaZda software before the parameters’ extraction, based on predefined settings.

Each examination was reviewed by a third researcher (P.A.S.), blinded to the final diagnosis, who performed the image segmentation. This step consisted of incorporating each lesion into a two-dimensional region of interest (ROI). A semi-automatic level-set technique was used for the definition and positioning of each ROI using geometry and gradient coordinates. When an incomplete overlap between the lesion and the ROI was observed, the researcher performed the necessary adjustments (Figure 1).

#### 2.4.2. Feature Extraction

The texture feature extraction from every ROI was automatically performed by the software. The resulted texture features were derived from the histogram analysis, gradient, run-length matrix (RLM), gray-level co-occurrence matrix (GLCM), autoregressive model, and wavelet transformation. The histogram analysis included the following nine parameters: mean, variance, skewness, kurtosis, and five percentiles (1%, 10%, 50%, 90%, and 99%). The gradient features were computed using 4 bits/pixel and included five parameters: absolute gradient mean, absolute gradient variance, absolute gradient skewness, absolute gradient kurtosis, and percentage of pixels with nonzero gradient. The RLM included five parameters (run-length nonuniformity, gray-level nonuniformity, long-run emphasis, short-run emphasis, and the fraction of image in runs). Each RLM parameter was computed using 6 bits/pixel and calculated for four directions on the image: vertical (V), horizontal (H), 135° (N), and 45° (Z) resulting in four variations of each feature. The eleven GLCM features were computed using 6 bits/pixel (angular second moment, contrast, correlation, the sum of squares, inverse difference moment, sum average, sum variance, sum entropy, entropy, difference variance, and difference entropy). Each COM parameter was calculated in four directions (V, H, N, and Z) for each inter-pixel distance (1, 2, 3, 4, and 5), resulting in 20 variations of each parameter. The autoregressive model included five parameters (θ 1–4 and σ). The wavelet transformation included a single parameter (wavelet energy) that was computed using 8 bits/pixel. This feature was calculated at six scales using four frequency bands (low–low, low–high, high–low, and high–high), resulting in 24 variations.

In total, 283 parameters were computed from every ROI. Two sets of measurements (lesion segmentation and feature extraction) were performed one week apart by the same researcher, and the resulted values were used to assess the intra-observer agreement. The intra-reader agreement was evaluated using the intraclass coefficient (IC) between the parameters extracted by the same researcher. Only the features that showed an IC of >85 were selected for further statistical analysis.

#### 2.4.3. Feature Selection

Two methods were used to ensure the selection of the most discriminative parameters. First, the MaZda software allows the selection of the best-suited parameters for differentiating between classes through several reduction methods. Three of them were applied, based on the probability of classification error and average correlation coefficients (POE + ACC), mutual information (MI), and Fisher coefficients (F, the ratio of between-class to within-class variance) [23], each of them providing a set of ten texture features.

Second, the parameters highlighted by the three methods underwent statistical analysis. The absolute values recorded by each of the previously selected parameters were compared between the two groups by conducting a univariate analysis test (the Mann–Whitney U test). The statistical significance level was set at a *p*-value of below 0.0019 after applying the Bonferroni correction (which implied dividing the standard 0.05 value to 26 variables; 23 were represented by the unique parameters provided by the reduction technique, one corresponding to the patients’ age, and two represented the major histopathological classes). Features that did not meet the above-mentioned criteria were excluded from further analysis.

#### 2.4.4. Class Prediction

Two methods were used to assess the selected features’ ability to distinguish between benign and malignant adnexal lesions. First, we investigated which of the parameters that showed statistically significant results at the univariate analysis are also independent predictors of malignancy. In this regard, a multiple regression analysis (using the “enter” input model) was conducted, with the computation of the coefficient of determination (R-squared) and the variance inflation factor (VIF). The “enter” input model included all variables that showed a *p*-value of below 0.05 and removed all variables that showed a *p*-value of more than 0.01. The residuals were tested for normal distribution by applying the D’Agostino–Pearson Test. Since a high VIF value is an indicator of multicollinearity, features that recorded a VIF of ≥10^4^ were excluded from further analysis. The predicted values were saved and subsequently used in a receiver operating characteristics (ROC) analysis to assess the diagnostic power of the entire prediction model. The ROC analysis was also used to determine the diagnostic power of features independently associated with ovarian malignancies, along with the calculation of the area under the curve (AUC), sensitivity, and specificity, with 95% confidence intervals (CIs). The ROC curves were calculated using the DeLong et al. method, and the binomial exact confidence intervals for the AUCs were reported. Optimal cut-off values were chosen using a common optimization step that maximized the Youden index for predicting patients with malignancies. Sensitivity (Se) and specificity (Sp) were computed from the same data, without further adjustments. Statistical analysis was performed using a commercially available dedicated software, MedCalc v14.8.1 (MedCalc Software, Mariakerke, Belgium).

Second, within the B11 program (part of the MaZda package), the use of textural features to differentiate between classes was further evaluated by the use of a classifier. The classifier chosen in this model was the k-nearest neighbor (KNN), which follows the partitioning method for clustering [24]. Two feature sets were subsequently imputed in the KNN. The first was the set composed of all the parameters highlighted by the selection methods (set 1), and the second was the set that contained the parameters that were demonstrated to be independently associated with the presence of malignant lesions (set 2). The classifier’s ability to distinguish between the two histological types of adnexal lesions was shown by quantifying its Se (true positive rate), Sp (true negative rate), and accuracy (Acc, percentage of correct classified lesions), with 95% CI. The entire workflow diagram is displayed in Figure 2.

## 3. Results

Of the 274 patients with adnexal lesions who were referred to our department during the study period, 120 (average age ± standard deviation: 38.15 ± 14.68 years; age range: 22–76 years) were included in this study after applying the inclusion and exclusion criteria. Patients were divided into benign (85 patients and 88 images) and malignant groups (35 patients and 35 images).

Twenty-three individual parameters were selected by applying the three reduction techniques. Three variations of the sum variance (SumVarnc) parameter (CH5D6SumVarnc, CH4D6SumVarnc, CH3D6SumVarnc) were selected by both Fisher and POE + ACC techniques, of which two were highlighted by all three methods (CH3D6SumVarnc and CH4D6SumVarnc), and two variations were selected only by the Fisher and mutual information techniques (CN2D6SumVarnc, CV4D6SumVarnc).

When comparing the absolute values, the Mann–Whitney U test showed statistically significant results in 18 of the 23 previously selected parameters. Five parameters were situated above the *p*-level threshold and therefore were excluded from further processing (WavEnHL_s-6/wavelet energy, *p* = 0.0283; ATeta4/parameter θ4, *p* = 0.0675; GD4Kurtosis/absolute gradient kurtosis, *p* = 0.0913; RZD6LngREmph, *p* = 0.3699; and WavEnLH_s-5, *p* = 0.014). The intra-rater agreement showed adequate reproducibility of all remaining texture parameters. The results of the univariate analysis and intra-reader agreement evaluation are displayed in Table 1.

The multivariate analysis showed a significance level of *p* < 0.0001, an R2 coefficient of determination of 0.4754, an adjusted R2 of 0.4153, and a multiple correlation coefficient of 0.6895. Three features were identified as independent predictors of malignant lesions (CH5D6SumOfSqs, CZ2D6SumVarnc, and CZ5D6SumOfSqs) (Table 2). Seven features were excluded from the model due to multicollinearity (as having a VIF >104) (CH3D6SumVarnc, CH4D6SumVarnc, CN2D6SumVarnc, CV3D6SumVarnc, CV4D6SumVarnc, CZ3D6SumOfSqs and CZ4D6SumOfSqs). The ROC analysis showed that the prediction model exceeded the diagnostic ability of all independent features in terms of both sensitivity and specificity (Table 3, Figure 3).

Two feature sets were analyzed separately by the KNN. The first (set 1) contained all 23 parameters initially selected by the reduction techniques, and the second set (set 2) contained only the three parameters that were independently associated with ovarian malignancies. Nineteen lesions were incorrectly classified by the KNN based on the first feature set and 18 lesions based on the second feature set. Nine lesions were misclassified by the KNN after the imputation of both sets: functional cyst (*n* = 1), hemorrhagic cyst (*n* = 1), endometrioma (*n* = 2), serous cystadenoma (*n* = 1), serous carcinoma (*n* = 4), mucinous carcinoma (*n* = 1). Additionally, the KNN based on the parameters from set 1 incorrectly classified: endometrioma (*n* = 1), serous cystadenoma (*n* = 2), mucinous cystadenoma (*n* = 1), serous carcinoma (*n* = 3), mucinous carcinoma (*n* = 1), endometroid carcinoma (*n* = 1). Separately, the KNN based on the parameters from set 2 misclassified: endometrioma (*n* = 1), serous cystadenoma (*n* = 3), ovarian abscess (*n* = 1), oophoritis (*n* = 1), serous carcinoma (*n* = 1), clear cell carcinoma (*n* = 1). The overall KNN performance and classification results are displayed in Table 4. The feature maps that display the image distribution of parameters across images belonging to the two groups are shown in Figure 4.

## 4. Discussion

### 4.1. Study Outcomes

Our results indicate that an important part of the initially selected parameters were variations of the GLCM-based features, namely SumVarnc and SumOfSqs (sum of squares). These variations showed similar results at the univariate analysis (having a *p*-value of less than 0.0001), a similar range of values for the benign (SumVarnc, 83.53–94.86; SumOfSqs, 25.03–91.34) and malignant groups (SumVarnc, 230.49–269.59; SumOfSqs, 62.24–69.91), and similar intra-rater reproducibility coefficients (SumVarnc, 0.95–0.96; SumOfSqs, 0.96).

The multivariate analysis indicated that three features were independently associated with the presence of ovarian malignancies (CH5D6SumOfSqs, CZ2D6SumVarnc, CZ5D6SumOfSqs). Of the independent parameters, the highest Se (90.48%; CI, 69.6–98.8%) and Sp (79.31%; CI, 69.3–87.3%) were once again achieved by GLCM-based features. Overall, all the features belonging to this class showed a high potential to discriminate between benign and malignant adnexal lesions, an observation that is in accordance with previous USTA research [7,12,15,16]. Moreover, it was demonstrated that GLCM parameters are uninfluenced by the variations of ROI depth and size, and gain settings, and also showed good operator repeatability and reproducibility [16].

Both the sum of squares and sum variance features reflect the degree of deviation from the mean gray level present in the ROI [25], and some authors even provide them with the same definition [26]. Sum variance quantifies the variance of the sum of gray levels (e.g., the spread in the sum of the gray-levels of pixel-pairs distribution) [27,28]. Within the GLCM, when the frequency of occurrence is equally concentrated in the lowest and highest cells of the matrix, this parameter’s values increase [29]. In other words, sum variance is a measure of heterogeneity that places higher weight on neighboring intensity level pairs that deviate more from the mean [30]. Similarly, the values of the sum of squares increase proportionally to the pixel gray value differences [27]. Our results show higher values of all variations of both parameters for the malignant compared to the benign group, and the difference between the measurements was statistically significant in all scenarios (*p* < 0.0001).

Several hypotheses can be formulated to explain the increased heterogeneity observed in the malignant group. A possible explanation is related to the fact that the morphological features (such as septations, solid areas within the cyst, wall structure, and papilations) were included within the ROI, thus creating a high contrast between fluid and solid components that was interpreted by the software as an expression of inhomogeneity. On the other hand, tissue heterogeneity is a well-known feature of malignancy, likely related to tumoral angiogenesis and cell infiltration [31,32], a feature that could be very well-reflected in the analyzed US images.

Another important aspect is related to the fluid component of the ovarian cystic lesions. It was previously documented that the fluid contained in these lesions express particularities for several histological groups, such as biomarkers [33,34], cellularity [35,36], and liquid properties [37,38]. Overall, the fluid contained in the malignant cystic lesions is more heterogeneous than one contained in benign lesions [37,39,40], which could also impact the USTA results. Moreover, we previously demonstrated that the texture analysis [41] and density measurements [42] of the fluid component can successfully discriminate alone between the two histopathological groups. Therefore, it is not possible to indicate exactly which of the lesions’ macro or microscopical components are responsible for the increased heterogeneity, and because of this, no direct link between texture parameters and the lesions’ appurtenance to a certain histopathological group can be stated. These matters were not addressed in previous USTA studies [7,11,12,13,14,15,16], although we consider them to be a key component in the interpretation of the TA results. In order to identify the tumoral components that directly impact the texture measurements, separate USTA of solid and liquid components would be required, under direct coordination with the pathology and surgical departments regarding the timing and handling of probes.

We were able to identify seven previously published papers that also followed the differentiation between benign and malignant adnexal masses based on USTA. Besides having similar backgrounds and aims, the workflow and the results were highly variable [7,11,12,13,14,15,16] (Table 5).

### 4.2. Study Population

Our study group included 123 images provided by 120 subjects. Similar research also included clustered observational data. Normally, analyses of data that include multiple observations per subject require a form of adjustment to account for the possible correlation between observations [43]. A single image corresponding to one lesion was retrieved from most of the subjects included in our study (*n* = 117). Two images, each corresponding to a different pathologically proven lesion, were retrieved from three patients. Since the images provided from the same patient were analyzed as separate entities, no statistical adjustments were performed. The same approach was followed by Aldahlawi et al. [15] (which included 169 images from 163 patients), and by Khazendar et al. [14] (which included 187 images from 177 subjects). On the other hand, the studies conducted by Acharya et al. [7,11,12,13] included between 100–130 images retrieved from the same lesion, also with no statistical adjustments regarding the reported diagnostic ability of the texture features.

### 4.3. Image Pre-Processing and Segmentation

In US images, the superposition of acoustical echoes with random phases and amplitudes produces speckle noise [44]. Also, variations in brightness could occur due to different operators and different settings used when acquiring the image [45]. The speckle noise and brightness variations inevitably have an impact on the measured texture parameters and therefore negatively influence the tissue comparability. For speckle noise reduction, we used a CNN-based approach [20]. Other studies used a slightly different modality to counteract these variations. Khazendar et al. [14] applied a noise reduction method in the pre-processing step based on nonlocal mean (NLM). However, recent publications demonstrated that deep learning algorithms (such as CNN) can easily outperform the NLM approach since it can automatically deal with both stationary and spatially varying noise patterns [46]. For brightness and contrast variations, we used the limitation of dynamics to mean and three standard deviations, which was successfully applied in previous TA studies based on US as well as other imaging modalities [45,47]. Aldahlawi et al. [15] also conducted the analysis based in MaZda based on BMP images, restrained from using any image correction methods. Also, in the researches conducted by Acharya et al. [7,11,13] the use of image correction methods was not mentioned.

Our segmentation process included a semi-automatic level set technique, and manual corrections were applied when necessary. In previous research with a similar workflow it was stated that this technique does not require intra- or inter-observer reproducibility assessment [48]. However, the researcher (P.A.S.) who conducted the segmentation process was asked to repeat the process one week later, to evaluate and ensure good intra-observer reproducibility of the texture measurements. Similar USTA research used an entirely manual ROI definition and the reproducibility was not assessed [14,15]. Two papers also implied a semi-automatic method [7,11], but also with no details regarding the agreement. One study [13] offered no information about lesion segmentation and ROI definition. However, it was documented that the ROI characteristics (such as delineation, size, shape, etc.) could influence the TA results [18], and therefore intra- or inter-observer reproducibility could be an important aspect.

### 4.4. Feature Extraction and Reduction

Feature extraction is the main step in the TA process and implies the computation of TA parameters from preselected regions. There are many methods from which the parameters can be retrieved, including statistical methods (GLCM, RLM, local binary patterns, etc), model-based methods (autoregressive and fractal models), transform methods (wavelet transformation), and many more, each including several specific categories of textural features [18]. In our study, 283 features belonging to six categories that were computed from every ROI. Most of the similar studies [11,14,15,16] extracted texture features belonging to two categories, two studies included three categories [7,12], and one study included five categories [13]. The GLCM-based features showed statistically significant and/or adequate classification results in four previous studies [7,13,15,16]. Although this category provided the highest discriminatory ability, it is possible that the use of multiple parameters from several texture methods could provide a more complete description of the image contents.

Three built-in feature reduction methods (Fisher, POE + ACC, and MI) were each used to select a set of features with high discriminatory ability. Of the two similar studies that were also conducted on the MaZda platform [15,16], none used any selection methods, and restrained form analyzing any other parameters than GLCM and wavelet-based features. The most used reduction technique in all previous studies was represented by the univariate analysis (Mann–Whitney U or Student *t*-test) [7,11,12,13,14,15,16]. Typically, a *p*-value of less than 0.05 is regarded as clinically significant [12], and this threshold was used in five previous researches [7,12,14,15,16]. Two studies [11,13] considered a *p*-value of less than “0.01 or 0.05” to be statistically significant. However, we emphasize that the statistical significance level needs to be adjusted, especially when tens or hundreds of parameters are computed within the same ROI.

### 4.5. Class Prediction

The KNN is one of the simplest classifiers [6]. Our choice for the KNN was determined by the fact that it showed high discriminatory power (100% sensitivity, specificity, positive predictive value, and accuracy) in a previous USTA research conducted by Acharya et al. [7], but also because it can be accessed through the B11 program (part of the MaZda package). This classifier performed poorly in the study conducted by Khazendar et al. [14] (Acc, 63–55%; Se, 55–71%; Sp, 49–49) but showed adequate classification abilities in a research performed by Acharya et al. [7] (100% average Acc, Se, and Sp). The very good results may be determined by the clustered input data, mostly since the second research [7] used 130 images from every lesion of the 20 enrolled subjects (50% with benign and 50% with malignant tumors). On the contrary, our study group, as well as the one used by Khazendar et al. [14] included fewer images obtained from a larger number of patients with a wider range of pathologies, which could represent a more accurate perspective of the capabilities of this tool.

A very important step in the classification process is splitting the study population into training and validation (testing) sets. Typically, approximately 70% of the acquired dataset is used for training and the remaining samples are used to evaluate the classifier’s performance [13]. We were unable to perform this task due to the limited number of ovarian malignancies (*n* = 35; 29.1% of the study population) that were referred to our center in the recruitment time that was approved by the ethical committee (16 months), and splitting this group into training (*n* = 24) and validation (*n* = 11) sets would not offer a fair assessment of the classifier’s performance. However, our classifier was also run two times, but with slightly different parameters. Nevertheless, the overall performance of our KNN tool was moderate at best, especially in terms of sensitivity of malignant lesions’ detection (71.43–80%). Overall, in previous studies, the highest diagnostic abilities were achieved on small study populations or few histopathological entities [7,11,12,13,16]. Thus, the USTA and CAD’s utility in the diagnosis of ovarian malignant lesions remains uncertain, and further studies are required to validate this method.

### 4.6. Future Perspectives

There is an undoubted need for new noninvasive methods that could correctly classify adnexal lesions, especially since the classic surgical and interventional methods for tissue sampling are invasive, offer inconsistent results, and expose the patients to a series of risks [49,50,51,52,53]. Also, the classic imaging assessment can be limited by the morphological features advocating for ovarian malignancy that appear in later stages of the disease, hence the need for developing early and precise biomarkers [54]. In this regard, TA can offer a non-invasive and objective description of the lesions’ contents, that could be further linked to the risk of malignant transformation of small/early detected adnexal lesions. Moreover, the TA could be integrated into prediction models along with other non-imaging parameters such as CA-125 levels, thus providing an individual malignancy risk assessment for every patient.

Previously, TA parameters and other radiomics models have been proved useful in predicting several oncologic phenotypic patterns [55], response to tumor treatment [56], and even the overall survival and metastasis risk [57]. If further research can demonstrate a direct link between imaging parameters and local malignant tissue characteristics, the USTA evaluation of adnexal malignancies could prove useful especially when biopsy can not be performed [58]. Moreover, if adnexal tumors’ genomics could be further linked to the dynamics of texture parameters, this approach could become a core component of personalized oncology [59]. It was previously demonstrated that MRI can augment the diagnosis in patients with indeterminate adnexal masses detected at TVUS [60]. Since MRI acquisitions can contain more information than USTA, it is possible that MRI-based TA could offer a more adequate characterization and discrimination of adnexal masses than USTA. Rongping et al. [61] proposed a combined model that integrated non-texture information (clinical and conventional MRI features) and texture features (extracted from T2-weighted imaging, diffusion-weighted imaging, and contrast-enhanced T1-weighted imaging) that was able to differentiate borderline epithelial ovarian tumors from FIGO stage I/II malignant epithelial ovarian tumors with 92.5% sensitivity, 86.4% specificity and an AUC of 0.962. Moreover, in a previously published study [41], we build a prediction model based on the texture features extracted from the fluid component of adnexal masses as seen on T2-weighted images that was able to discriminate benign from malignant lesions with 84.62% sensitivity, 80% specificity, and an AUC of 0.841. Therefore, a direct comparison between US and MRI-derived texture features extracted from the same cohort could be useful to establish which of the two methods can offer better diagnostic rates in terms of adnexal lesions.

### 4.7. Study Limitations

Our study had several limitations. Due to its retrospective nature, it may contain selection and verification bias regarding the gynecological follow-up and management, which mainly depend on the referral hospital and the status of the institution. The study population included almost four times more benign than malignant lesions, therefore not allowing the data to be split into training and validation sets. The menstrual phase and menopausal status were not considered since it is was inconsistently mentioned in the retrieved medical data. The fact that two researchers were aware of the final diagnosis of the lesions can also be considered a limitation. However, because at the time of the US examination several patients exhibited multiple adnexal lesions, this approach was necessary for selecting only documented lesions. After this step, the two researchers were not involved in the processes of image segmentation, statistical analysis, or reporting the results. The software used in this study can be regarded as outdated since an official new version has not been released in more than ten years. However, for this research, we used a newly developed beta version released four years ago (MaZda version 5). Although currently several other texture programs have been developed, few others are able to offer built-in techniques for feature reduction and vector classification within an intuitive interface that can be used by non-image processing specialists, such as medical doctors. Considering these limitations and also the pilot nature of some of the previously published studies, the USTA approach for discriminating adnexal lesions requires prospective research for both validation and establishment of its clinical utility compared to the classic imaging methods.

## 5. Conclusions

We demonstrated a statistically significant difference between adnexal benign and malignant features based on ultrasound-derived texture features. Although successful, it is unclear whether these texture features reflect the lesions’ appurtenance to a certain histopathological group or the ultrasonographic differences between the fluid and the solid components. Further studies are required to identify the exact substrate that determines textural differentiation.

## Figures and Tables

**Figure 1 diagnostics-11-00812-f001:**
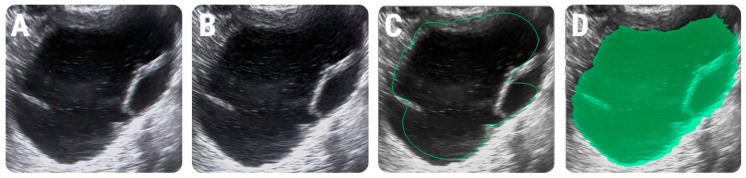
Image pre-processing workflow and the positioning of the region of interest (ROI). (**A**) The ultrasound image of a 34-year-old patient with histologically-proved mucinous cystadenoma that underwent speckle-noise reduction (**B**); the initial ROI that was automatically delineated by the software (green line) (**C**) and the final ROI after manual adjustments (green) (**D**).

**Figure 2 diagnostics-11-00812-f002:**
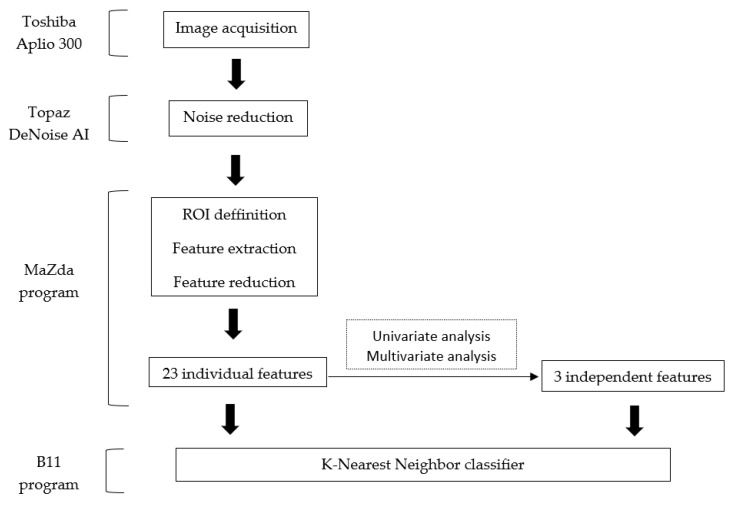
Workflow diagram. ROI, region of interest.

**Figure 3 diagnostics-11-00812-f003:**
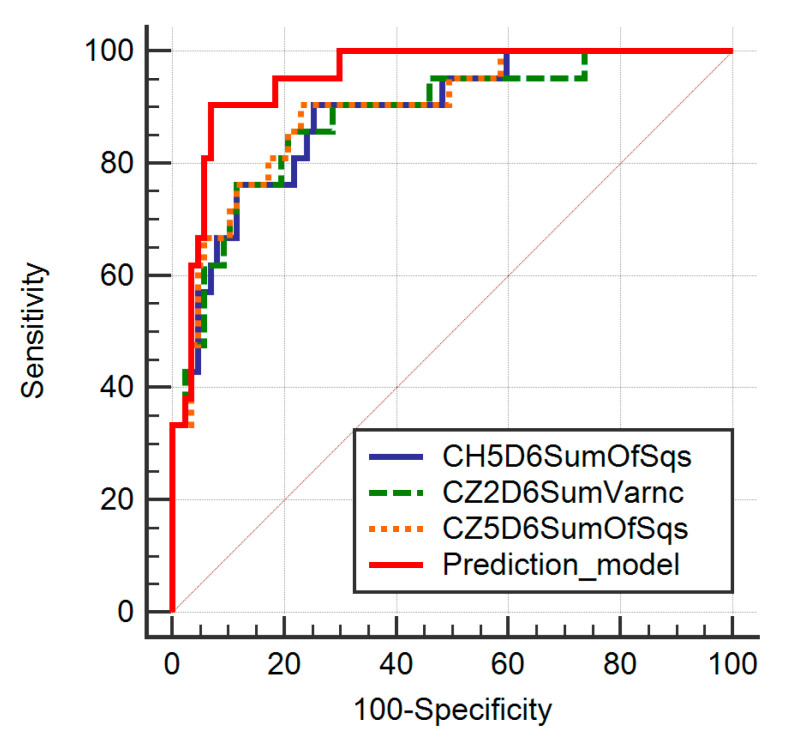
Receiver operating characteristic curve of the four texture parameters independently associated with the presence of malignant lesions and the prediction model. SumVarnc, sum variance; SumOfSqs, sum of squares.

**Figure 4 diagnostics-11-00812-f004:**
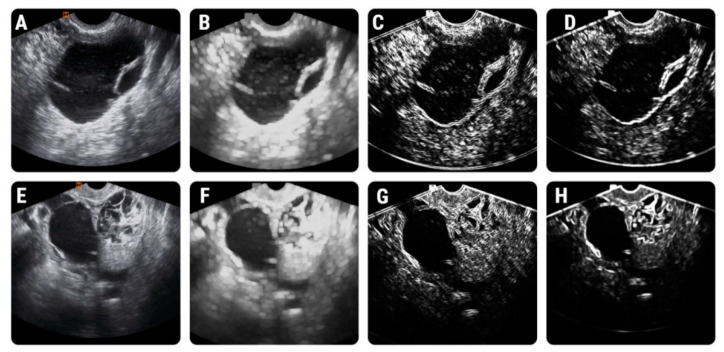
Texture maps that show the distribution of texture features in selected images. (**A**) The ultrasound image of a 34-year-old patient with histologically-proved mucinous cystadenoma; (**B**–**D**) show the distribution of Perc99, CV5D6SumVarnc, and CV5D6SumOfSqs in image (**A**); (**E**) The ultrasound image of a 67-year-old patient with histologically-proved serous ovarian carcinoma; (**F**–**H**) show the distribution of Perc99, CV5D6SumVarnc, and CV5D6SumOfSqs in image (**E**).

**Table 1 diagnostics-11-00812-t001:** The univariate analysis (Mann–Whitney U test) and the intra-reader agreement evaluation results.

Parameter	*p*-Value	Benign Group		Malignant Group		Agreement	
		Median	IQR	Median	IQR	ICC	95% CI
**Fisher**							
CH4D6SumVarnc	**<0.0001**	85.88	59.73–131.59	242.63	167.91–455.47	0.96	0.94–0.97
CH3D6SumVarnc	**<0.0001**	88.27	62.54–136.3	258.28	172.04–463.22	0.96	0.94–0.97
CH5D6SumVarnc	**<0.0001**	84.31	56.33–128.25	235.9	164.45–450.94	0.95	0.94–0.97
CV5D6SumVarnc	**<0.0001**	83.53	55.74–127.05	230.49	143.38–438.08	0.96	0.94–0.97
CV4D6SumVarnc	**<0.0001**	85.37	56.91–132.69	234.69	148.02–449.39	0.96	0.94–0.97
CV3D6SumVarnc	**<0.0001**	86.71	59.6–139.33	238.15	154.47–465.49	0.96	0.94–0.97
CH2D6SumVarnc	**<0.0001**	94.86	67.57–148.25	269.59	180.29–480.45	0.96	0.94–0.97
CN2D6SumVarnc	**<0.0001**	89.56	61.13–136.95	247.13	158.06–464.65	0.96	0.94–0.97
CN3D6SumVarnc	**<0.0001**	83.48	58.51–129.63	236.66	147.31–449.5	0.96	0.94–0.97
CV2D6SumVarnc	**<0.0001**	91.49	63.79–147.72	251.8	168.09–479.01	0.96	0.94–0.97
**POE + ACC**							
WavEnHL_s-6	0.0283	108.02	54.64–173.56	124.12	110.04–214.94	0.99	0.99–0.99
Kurtosis	**0.0005**	10.27	4.55–21.56	4.07	1.12–7.23	0.92	0.89–0.94
ATeta4	0.0675	0.18	0.09–0.24	0.14	0.08–0.16	0.98	0.97–0.98
GD4Kurtosis	0.0913	50.09	14.94–68.34	13.01	4.–44.66	0.99	0.99–0.99
RZD6LngREmph	0.3699	3.4	2.25–9.54	3.09	2.17–4.72	0.97	0.95–0.98
Perc99	**<0.0001**	116	85.5–144	166	150–207.25	0.93	0.9–0.95
WavEnLH_s-5	0.014	92.19	63.68–138.44	122.81	101.37–153.98	0.98	0.97–0.98
**Mutual Information**							
CZ4D6SumOfSqs	**<0.0001**	25.24	17.23–39.44	64.54	48.28–118.29	0.96	0.95–0.97
CZ5D6SumOfSqs	**<0.0001**	25.03	17.14–38.63	62.24	48.03–116.66	0.96	0.95–0.97
CH5D6SumOfSqs	**<0.0001**	25.13	16.93–40	67.71	48.79–117.8	0.96	0.94–0.97
CZ2D6SumOfSqs	**<0.0001**	25.6	17.85–41.33	69.91	49.11–122.1	0.96	0.94–0.97
CZ3D6SumOfSqs	**<0.0001**	25.51	17.48–40.51	67.11	48.76–120.45	0.96	0.94–0.97
CZ2D6SumVarnc	**<0.0001**	91.34	61.89–142.25	240.57	169.41–470.34	0.96	0.94–0.97

Values in bold are statistically significant. IQR, interquartile range; POE + ACC, probability of classification error and average correlation coefficient; ICC, intraclass coefficient; SumVarnc, sum variance; WavEnHL_s-6, wavelet energy; GD4Kurtosis, kurtosis; CV5D6SumEntrp, sum entropy; ATeta4, parameter θ4; Perc99, 99% percentile; ASigma, parameter σ; SumOfSqs, sum of squares.

**Table 2 diagnostics-11-00812-t002:** Multivariate analysis results showing the parameters independently associated with the presence of malignant lesions. Bold values are statistically significant (*p* < 0.05). VIF, variance inflation factor.

Parameter	Coefficient	Standard Error	*p*-Value	VIF
CH2D6SumVarnc	0.015	0.013	0.2341	3718.896
CH5D6SumOfSqs	−0.108	0.045	**0.019**	2905.638
CH5D6SumVarnc	0.013	0.009	0.1665	1629.273
CN3D6SumVarnc	−0.009	0.008	0.2547	1396.956
CV2D6SumVarnc	0.0194	0.014	0.175	4275.282
CV5D6SumVarnc	−0.0017	0.009	0.859	1769.967
CZ2D6SumOfSqs	−0.0287	0.059	0.63	4969.092
CZ2D6SumVarnc	−0.0246	0.009	**0.011**	1880.17
CZ5D6SumOfSqs	0.097	0.039	**0.014**	2058.43
Kurtosis	<0.001	0.002	0.8365	1.663
Perc99	0.001	0.001	0.5304	8.415

**Table 3 diagnostics-11-00812-t003:** The receiver operating characteristic analysis results of the parameters that are independently associated with the presence of ovarian malignancy and the prediction model consisting of these parameters. Between the brackets are the values corresponding to the 95% confidence interval.

Parameter	AUC	Significance Level	J	Cut-Off	Se (%)	Sp (%)
CH5D6SumOfSqs	0.887 (0.812–0.94)	<0.0001	0.65	>39.77	85.71 (63.7–97)	74.71 (64.3–83.4)
CZ2D6SumVarnc	0.883 (0.807–0.937)	<0.0001	0.65	>151.46	85.71 (63.7–97)	79.31 (69.3–87.3)
CZ5D6SumOfSqs	0.895 (0.821–0.946)	<0.0001	0.67	>38.77	90.48 (69.6–98.8)	77.01 (66.8–85.4)
Prediction model	0.951 (0.891–0.983)	<0.0001	0.83	>0.31	90.48 (69.6–98.8)	93.1 (85.6–97.4)

SumVarnc, sum variance; SumOfSqs, sum of squares; Perc 99, 99th percentile.

**Table 4 diagnostics-11-00812-t004:** The k-nearest neighbor classifier’s results. The values between the brackets corresponding to the 95% confidence interval. KNN, k-nearest neighbor classifier.

Input Parameters	Set 1	Set 2
Accuracy (%)	84.55 (76.93–90.44)	85.37 (77.86–91.09)
Sensitivity (%)	71.43 (53.7–85.36)	80 (63.06–91.56)
Specificity (%)	89.77 (81.47–95.22)	87.5 (78.73–93.59)
Positive Predictive Value (%)	73.53 (59.1–84.23)	71.79 (58.84–81.93)
Negative Predictive Value (%)	88.76 (82.32–93.06)	91.67 (84.95–95.54)
**Study population**		
benign group (*n* = 88)	9	11
functional cyst (*n* = 7)	1	1
hemorrhagic cyst (*n* = 5)	1	1
endometrioma (*n* = 28)	3	3
serous cystadenoma (*n* = 26)	3	4
mesothelial inclusion cyst (*n* = 2)	-	-
mucinous cystadenoma (*n* = 6)	1	-
ovarian abscess (*n* = 6)	-	1
oophoritis (*n* = 2)	-	1
teratoma (*n* = 6)	-	
malignant group (*n* = 35)	10	7
serous carcinoma (*n* = 24)	7	5
endometroid carcinoma (*n* = 2)	1	-
mucinous carcinoma (*n* = 6)	2	1
clear cell carcinoma (*n* = 3)	-	1

“-” corresponds to no lesion from a histopathological group being misclassified.

**Table 5 diagnostics-11-00812-t005:** Summary of benign and malignant ovarian masses classification using texture features from ultrasound images.

Author, Year	Study Group	Texture Features	Classifier	Performance		
				Acc (%)	Se (%)	Sp (%)
Acharya et al. 2013 [11]	n_s_ = 20	LBP, LTE	SVM	99.9	100	99.8
	n_i_ = 2000					
Acharya et al. 2012 [12]	n_s_ = 20	Hu i.m., Gabor, Entropies	PNN	99.8S	99.2	99.6
	n_i_ = 2600					
Acharya et al. 2012 [13]	n_s_ = 20	SD, FD,	DT	97	94.3	99.7
	n_i_ = 2000	GLCM, RLM, HOS				
Acharya et al. 2014 [7]	n_s_ = 20; n_i_ = 2600	FOS, GLCM, RLM	SVM	84.7–100	81–100	88.46–100
			DT	98.54	98.15	98.92
			KNN	100	100	100
			NB	67.35	60.62	74.08
			PNN	100	100	100
Khazendar et al. 2015 [14]	n_s_ = 187	FOS, LBP	KNN	63–55	55–71	49–69
	n_i_ = 177					
Aldahlawi et al. 2017 [15]	n_s_ = 163;	GLCM	-	-	71–75	55–60
	n_i_ = 169	Wavelet	-	-	50–62	46–60
Hamid. 2011 [16]	n_s_ = 20	GLCM	-	-	100	90
	n_i_ = 20	Wavelet	-	-	100	90

n_s_, number of subjects; ni, number of analyzed images; acc, accuracy; Se, sensibility; Sp, specificity; LBP, Local Binary Patterns; LTE, Laws Texture Energy; SVM, Support Vector Machine; Hu i.m., Hu’s invariant moments; PNN, probabilistic neural network; DT, Decision Tree classifier; SD, standard deviation; FD, fractal dimension; GLCM, gray-level co-occurrence matrix; RLM, run-length matrix; HOS, higher-order spectra; FOS, first-order statistics; KNN, k-Nearest Neighbor; NB, naïve Bayes; -, no information regarding this aspect could be found in the study.

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
