# Peer review of "Ultrasonography in the Diagnosis of Adnexal Lesions: The Role of Texture Analysis"

_diagnostics, 2021, doi:10.3390/diagnostics11050812_

Round 1
Reviewer 1 Report
It is my pleasure to have the opportunity to read your manuscript. It's an interesting topic for ultrasonography in the diagnosis of adnexal Lesions. However, although the analytical methods in this study are complex for materials and methods. But the study population is too small for training and validation as the authors mentioned in their study limitations. All the results need to be further validated when the study case number comes to enough.
Reviewer 2 Report
Ultrasonography-based texture analysis (USTA) is an extremely interesting Computer-Aided Diagnosis (CAD) technique; it reminds me how technology is progressively invading the role of physicians.
I only have a few suggestions to give:
-In the abstract you conclude with this sentence: “contrary to previous research, the USTA, although statistically significant, is insufficient to certify the histological appurtenance of adnexal lesions” but in the conclusions section of the manuscript you state “We demonstrated a statistically significant difference between adnexal benign and 522 malignant features based on ultrasound-derived texture features”. I exactly know what you mean because I attentively read all the article, but I think you should write better this concept for a quick reader; it seems you disagree with yourselves (again I know you do not).
-somewhere in the discussion, perhaps in the “future perspectives”, you should mention something about MRI; MRI is, in fact, crucial in the evaluation of adnexal masses; maybe, in the future, USTA should be compares to MRI results (idea for a future study?);
-the manuscript should be revised by an English proficient professional, only to improve a few expressions.
Round 2
Reviewer 1 Report
Although the authors made some revisions for this paper, still need more data set for training. I will keep my previous comment. They need more study cases to support their analysis result.
Author Response
Dear Reviewer,
Thank you very much for the observation. The KNN is considered one of the simple classifiers (doi: 10.1177/1533034614547445) which can perform well even with lesser data (doi: 10.7785/tcrtexpress.2013.600273). This classifier was also used in two of the previously published USTA studies, with 20 and 187 patients, respectively, and with very different outcomes (doi.org/10.7785/tcrtexpress.2013.60027 and orca.cf.ac.uk/14788/).
As we mentioned before, our study's first aim was to validate the texture workflow that was presented by previous research as successful, and did not focus to provide a stand-alone trained and tested machine learning algorithm.